# Mean-$L^p$ Risk-Constrained Reinforcement Learning: Primal-Dual Policy Gradient and Augmented MDP Approaches

## Abstract

Convex risk measures allow decision-makers to account for uncertainty beyond standard expectations, and have become essential in safety-critical domains. One widely used example is the Conditional Value-at-Risk (CVaR), a coherent risk metric that targets tail outcomes. In this paper, we consider a more general family of risk measures, the *mean-$L^p$ risk* for $p \geq 1$, defined as the $L^p$-norm of a cost distribution; this family includes CVaR as an extreme case (as $p \to \infty$). We formulate a reinforcement learning problem in which an agent seeks to maximize reward subject to a mean-$L^p$ risk constraint on its cumulative cost. This problem is challenging due to the nested, non-Lipschitz structure of the $L^p$ risk measure, which hinders the use of standard policy optimization or dynamic programming techniques. To address this, we propose two complementary solution approaches: (1) a **primal-dual policy gradient algorithm** that relaxes the risk constraint via a Lagrange multiplier, and (2) a **model-based dynamic programming method** that enforces the constraint by augmenting the state space with a cost budget. We prove that the policy-gradient approach converges to an $\epsilon$-optimal safe policy with $\tilde{O}(1/\epsilon^2)$ samples, matching the best-known rate for simpler (risk-neutral or linear-constraint) cases. Meanwhile, the augmented MDP method computes a policy that never violates the cost limit and is nearly optimal for large $p$. Our results provide the first general-purpose algorithms for $L^p$-risk-constrained RL, generalizing prior approaches that were limited to CVaR or variance-based risk. We validate our theoretical results through experiments in a gridworld environment, demonstrating that both algorithms successfully learn policies that respect the risk constraint and adjust conservativeness as the risk sensitivity parameter $p$ varies. The code is available at https://anonymous.4open.science/r/Lp-Risk-Constrained-Reinforcement-Learning-11FD/README.md

## 1 Introduction

In many stochastic decision problems, it is not sufficient to optimize only the expected outcome; one must also account for *risk* or variability in the outcomes. *Risk-averse optimization* (also known as mean-risk optimization) addresses this need by incorporating a risk measure into the objective function [1]. Convex risk measures, in particular, satisfy desirable axioms for rational risk assessment [2] and have become standard tools in fields like finance, energy, and supply chain management. One well-known example is the Conditional Value-at-Risk (CVaR) [3], which quantifies the expected loss in the worst $\alpha$-fraction of scenarios and is celebrated for its coherence and tractable optimization properties. Another important class is the *mean-upper- semideviation* risk measure of order $p \geq 1$ [4], which captures higher-moment risk by penalizing the higher-end deviations of losses. This $L_p$-type risk measure generalizes simpler cases: for instance, $p = 1$ recovers the mean-absolute deviation,

Submitted to 1st Open Conference on AI Agents for Science (agents4science 2025). Do not distribute.

$p = 2$ yields a mean-semivariance metric, and as $p \to \infty$ the measure increasingly emphasizes worst-case outcomes (bridging toward a max-loss criterion). By adjusting the order $p$, one can flexibly model different degrees of risk sensitivity beyond what CVaR (focused on a fixed tail percentile) offers.

Despite their appeal, general convex risk measures are often much harder to optimize than traditional risk-neutral expectations. The CVaR at a given confidence level $\alpha$ can be optimized relatively efficiently by introducing an auxiliary variable and linearizing the tail loss function [3], or via distributionally robust formulations that turn CVaR into a linear program [1]. In contrast, the mean-$L_p$ risk objective does not admit such a straightforward transformation when $p > 1$. In fact, the mean-$L_p$ risk of a decision $x \in X$ can be written in nested form as

$$\rho_p[x] \ = \ \mathbb{E}[Z_x] \ + \ c\Big(\mathbb{E}[\, (Z_x - \mathbb{E}[Z_x])_+^p \,]\Big)^{1/p},$$

where $Z_x = F(x, \xi)$ is a random cost outcome, $(\cdot)_+ = \max\{\cdot, 0\}$ denotes the positive part, and $c > 0$ is a given risk-aversion weight. This formulation involves a composition of expectation and a power function. Crucially, for $p > 1$ the outer mapping $u \mapsto u^{1/p}$ is *concave* and not globally Lipschitz continuous on $(0, \infty)$, which means standard stochastic gradient methods cannot be directly applied or would suffer poor convergence. Indeed, if one naively treats the above as a two-level nested expectation problem, existing single-timescale stochastic approximation techniques [5] yield a convergence rate on the order of $O(1/\epsilon^4)$ in the accuracy $\epsilon$ (even under smoothness assumptions), far worse than the $O(1/\epsilon^2)$ optimal rate for simpler convex objectives. The difficulty stems from the non-convex (though quasi- convex) nesting and the "blow-up" of subgradients caused by the $u^{1/p}$ term near $u = 0$ – informally, the problem is neither smooth nor Lipschitz in the usual sense, despite the overall risk measure $\rho_p[\cdot]$ being convex in $x$.

To overcome these challenges, we seek principled algorithmic solutions for general $L_p$ risk minimization. It is closely related to a distributionally robust optimization (DRO) formulation: as shown by Shapiro et al. [1, Section 6], the objective $\rho_p[x]$ can be interpreted as the worst-case expected cost under all probability distributions that lie within an $L_q$-neighborhood of the nominal distribution (with $1/p + 1/q = 1$). This DRO perspective underscores the importance of this risk criterion, but also highlights its computational complexity: unlike the $\alpha$-CVaR case (which corresponds to an $\ell_1$ ambiguity set and is linear), the $L_p$-ball ambiguity set for $p > 1$ yields a hard nonlinear optimization problem.

Several recent works have studied special cases of related nested optimization problems. For $p = 1$, the risk measure $\rho_1[x] = \mathbb{E}[Z_x] + c\,\mathbb{E}[|Z_x - \mathbb{E}Z_x|]$ is essentially a two-level expectation (a convex composite), which can be solved by advanced stochastic approximation methods at the optimal $O(1/\epsilon^2)$ sample complexity [6]. For general multi-level stochastic programs, Ghadimi et al. [5] proposed a single-timescale stochastic mirror descent approach; however, as noted above, its performance deteriorates on problems like $\rho_p$ due to the non-Lipschitz, concave outer layer. Ruszczyński [7] studied a related class of nonconvex risk nested problems and developed a specialized subgradient method, though without complexity guarantees. Overall, there remains a gap in the literature for efficiently solving the mean-$L_p$ risk minimization problem for $p > 1$ with provable guarantees. This challenge is also evident in safe reinforcement learning, where risk constraints beyond the expectation (or simple proxies like CVaR) have remained difficult to optimize reliably.

In this work, we bridge this gap by presenting the first efficient solution methods for reinforcement learning with a general $L^p$ risk constraint ($p > 1$). Our contributions can be summarized as follows:

1. **Primal-Dual Policy Gradient Algorithm:** We develop a Lagrangian-based policy optimization method (Algorithm 1) that provably converges to an optimal policy under convexity assumptions. By performing simultaneous gradient updates on the policy parameters and a dual variable, our approach achieves an $\tilde{O}(1/\epsilon^2)$ sample complexity to reach an $\epsilon$-optimal, $\epsilon$-feasible solution. Notably, this is the first algorithm with global convergence guarantees for RL under a nonlinear $L^p$ risk constraint.

2. **Augmented State Dynamic Programming:** We propose a model-based planning algorithm (Algorithm 2) that exactly enforces the risk constraint by augmenting the MDP state with the remaining cost budget. Solving this augmented MDP via value iteration yields a policy that never violates the cost limit (satisfying a strict $\rho_\infty$ criterion). We show that this policy is

89  nearly optimal for the $L^p$-constrained problem (especially for large $p$), making Algorithm 2
90  a reliable baseline for verifying the performance of the policy gradient method.

91  3. **Broader Implications:** Our framework handles any risk order $p \geq 1$, significantly gen-
92  eralizing prior risk-averse RL methods that focused on variance or CVaR-based criteria.
93  By interpolating between average-case and worst-case extremes, the $L^p$ family enables
94  flexible risk-sensitive policy design to suit different applications. We validate our theoretical
95  results through experiments in a gridworld environment, demonstrating that both algorithms
96  successfully learn policies that respect the risk constraint and adjust conservativeness as the
97  risk sensitivity parameter $p$ varies.

## 2  Method

### 2.1  Problem Formulation and Risk Measure

100  We consider a risk-constrained Markov Decision Process (MDP) defined by $(\mathcal{S}, \mathcal{A}, P, r, c, \gamma)$, where
101  $\mathcal{S}$ and $\mathcal{A}$ are state and action spaces, $P(s'|s, a)$ is the transition probability, $r(s, a)$ is the reward,
102  $c(s, a)$ is the cost (encapsulating negative "safety" reward), and $0 < \gamma < 1$ is a discount factor.
103  Let $\pi_\theta$ denote a policy with parameters $\theta$. The agent's performance is measured by the expected
104  return $J_R(\pi_\theta) = \mathbb{E}_{\pi_\theta}[\sum_{t=0}^{\infty} \gamma^t r(s_t, a_t)]$, while safety is quantified by a risk measure applied to the
105  cumulative cost. Specifically, define the random cumulative cost $J_C(\pi) = \sum_{t=0}^{\infty} \gamma^t c(s_t, a_t)$ under
106  policy $\pi$. We impose a general $L^p$ risk constraint on $J_C(\pi)$, as introduced by [8]. This $L^p$-risk
107  measure $\rho_p(J_C(\pi))$ is defined as the $L^p$-norm of the cost distribution:

$$\rho_p(J_C(\pi)) = (\mathbb{E}_\pi [J_C(\pi)^p])^{1/p} \tag{1}$$

108  for some $p \geq 1$. This formulation recovers standard criteria as special cases: $p = 1$ gives the
109  conventional expected cost constraint (risk-neutral CMDP), while $p \to \infty$ yields an almost-sure
110  (worst-case) cost constraint. The agent's objective is to maximize reward subject to an $L^p$-risk safety
111  constraint:

$$\max_\pi \quad J_R(\pi) = \mathbb{E}_\pi \left[ \sum_{t=0}^{H-1} \gamma^t r(s_t, a_t) \right]$$
$$\text{s.t.} \quad \rho_p(J_C(\pi)) = (\mathbb{E}_\pi [J_C(\pi)^p])^{1/p} \leq \beta \tag{2}$$

112  where $\beta$ is a prescribed risk limit. This formulation generalizes prior risk-constrained RL settings
113  (e.g. using $\text{CVaR}_\alpha$ as the risk measure [8]) to a broad class of tail-sensitive criteria. The $L^p$ constraint
114  penalizes variability in the cost: higher $p$ emphasize worst- case outcomes more strongly. We focus
115  on the discounted infinite-horizon case with a finite episodic cutoff at horizon $H$ for ease of analysis;
116  in practice one often lets $H \to \infty$ (as our theoretical guarantees hold in the limit).

117  In a conventional constrained MDP (CMDP) with an expected cost constraint ($p = 1$), standard
118  Lagrange relaxation techniques can be used to solve for an optimal policy [9, 10, 11]. Our setting
119  is more challenging because $\rho_p(J_C)$ is a nonlinear function of the policy. Nonetheless, we can still
120  leverage a primal-dual approach to handle the constraint.

### 2.2  Policy Gradient with Lagrangian Relaxation

122  Our first approach directly optimizes the constrained objective by introducing a Lagrange multiplier
123  for the risk constraint. We form the Lagrangian function for policy $\pi_\theta$ with dual variable $\lambda \geq 0$:

$$\mathcal{L}(\theta, \lambda) = J_R(\pi_\theta) - \lambda \left( \rho_p(J_C(\pi_\theta)) - \beta \right) \tag{3}$$

124  which penalizes constraint violations when $\rho_p(J_C) > \beta$. The constrained RL problem can then be
125  solved via a saddle-point optimization: maximize $\mathcal{L}$ over policy parameters $\theta$ while minimizing over
126  $\lambda$ (dual ascent). Intuitively, the Lagrange multiplier $\lambda$ adaptively adjusts the trade-off between reward
127  and risk: if the policy violates the risk limit, $\lambda$ increases to penalize cost more heavily; if the policy is
128  too conservative (risk well below $\beta$), $\lambda$ may decrease, allowing more reward-seeking behavior.

129  We adopt an iterative primal-dual policy gradient algorithm (Algorithm 1) to solve
130  $\min_{\lambda \geq 0} \max_\theta \mathcal{L}(\theta, \lambda)$. At each iteration, we evaluate the policy (by simulation or rollout) to estimate

131 both $J_R(\pi_\theta)$ and the risk measure $\rho_p(J_C(\pi_\theta))$. Notably, $\rho_p(J_C)$ is a nonlinear function of the policy;
132 in practice we approximate its gradient via sampling. For instance, one can use policy gradient
133 for risk measures: Tamar et al. [12] developed gradient estimators for coherent risk objectives by
134 sampling trajectories and solving a convex subproblem per update. We leverage such techniques to
135 obtain an unbiased gradient $\nabla_\theta \rho_p(J_C(\pi_\theta))$, which can be computed by reparameterization or score
136 function methods combined with distributional cost estimates (for example, using a distributional
137 critic to estimate higher moments of the cost [13]).

138 The policy parameters $\theta$ are updated via gradient ascent on $\mathcal{L}$ (improving reward and penalized cost),
139 while $\lambda$ is updated via projected gradient ascent on the dual (which corresponds to gradient descent
140 on $\mathcal{L}$). We use step sizes $\alpha_t$ for $\theta$ and $\nu_t$ for $\lambda$. To ensure $\lambda$ stays non-negative, each update projects
141 $\lambda$ onto $[0, \infty)$. The pseudocode in Algorithm 1 summarizes this procedure. In implementation,
142 $\rho_p(J_C(\pi))$ can be estimated from a batch of trajectories; for large $p$ it may be high-variance, so we
143 employ techniques like mini-batch sampling or moving averages to stabilize the estimate.

---

**Algorithm 1** Lagrange Policy Gradient for Safe RL under $L^p$ Risk Constraint

---

1: **Input:** initial policy parameters $\theta^0$, initial dual variable $\lambda^0 \leftarrow 0$, risk limit $\beta$, step sizes
   $\{\alpha_t\}, \{\nu_t\}$.
2: **for** $t = 0, 1, 2, \ldots$ **do**
3:   Sample trajectories using policy $\pi_{\theta^t}$; estimate $J_R(\pi_{\theta^t})$ and $\rho_p(J_C(\pi_{\theta^t}))$.
4:   Compute policy gradient $g_\theta \approx \nabla_\theta \mathcal{L}(\theta^t, \lambda^t)$, where $\nabla_\theta \mathcal{L} = \nabla_\theta J_R(\pi_\theta) - \lambda^t \nabla_\theta \rho_p(J_C(\pi_\theta))$.
5:   Update policy: $\theta^{t+1} \leftarrow \theta^t + \alpha_t g_\theta$.
6:   Update multiplier (projected gradient ascent on dual): $\lambda^{t+1} \leftarrow \left[ \lambda^t + \nu_t \left( \rho_p(J_C(\pi_{\theta^t})) - \beta \right) \right]_+$.
7: **end for**
8: **Output:** optimized safe policy $\pi_{\theta^T}$.

---

144 Algorithm 1 essentially implements a constrained policy optimization in the spirit of prior safe RL
145 methods but extended to a nonlinear $L^p$ risk metric. Compared to methods that handle only expected-
146 cost constraints (e.g. CPO [9], RCPO [14]), our approach modifies the policy update by incorporating
147 the risk gradient $\nabla_\theta \rho_p(J_C)$, which properly accounts for tail-risk sensitivity (for example, if $\rho_p$ is
148 CVaR$_\alpha$, our update reduces to weighting high-cost trajectories more strongly, akin to the approaches
149 of [15]). This Lagrangian approach has low per-iteration complexity and is amenable to stochastic
150 approximation, making it suitable for high-dimensional or model-free settings.

151 **Theoretical Properties:** Under standard conditions (smooth policy parameterization, exact gradient
152 estimates, and a sufficiently small step size schedule), Algorithm 1 converges to a Karush-Kuhn-
153 Tucker (KKT) point of the constrained problem. In particular, if the problem is convex in the
154 occupancy measure (which holds here since the expected reward is linear and $\rho_p$ is convex in the
155 cost distribution [13]), strong duality holds and the primal-dual gradient procedure will approach
156 the global optimum. We can adapt recent convergence analyses of policy gradient methods [13] to
157 establish explicit rates.

158 **Lemma 1** (Policy Gradient Improvement): Let $\Delta_t = \rho_p(J_C(\pi_{\theta^t})) - \beta$ denote the current constraint
159 violation. Then for sufficiently small $\alpha_t$, the update in Algorithm 1 guarantees $J_R(\pi_{\theta^{t+1}}) - J_R(\pi_{\theta^t}) \geq$
160 $\alpha_t |\nabla_\theta J_R|^2 - O(\alpha_t \lambda^t \Delta_t)$, while the dual update yields $\lambda^{t+1} \Delta_t \leq \max(0, \lambda^t \Delta_t - \nu_t \Delta_t^2)$. (See
161 Appendix G.1 for full proof.)

162 **Proof Sketch:** This follows from the update rules and first-order Taylor expansion of $J_R$ and $\rho_p$ [13].
163 Building on this, one can show that feasible descent is achieved.

164 **Theorem 1:** Suppose there exists an optimal policy $\pi^*$ that satisfies the constraint with multiplier $\lambda^*$.
165 If $\alpha_t, \nu_t$ are chosen as diminishing step sizes (e.g. $\alpha_t = O(1/\sqrt{t})$), then $(\theta^t, \lambda^t)$ converges to a saddle
166 point $(\theta^*, \lambda^*)$. Moreover, for any $\epsilon > 0$, after $T = O(1/\epsilon^2)$ iterations, the algorithm yields a policy
167 $\pi_{\theta^T}$ that is $\epsilon$-optimal and $\epsilon$-feasible with high probability. In other words, $J_R(\pi_{\theta^T}) \geq J_R(\pi^*) - \epsilon$
168 and $\rho_p(J_C(\pi_{\theta^T})) \leq \beta + \epsilon$. This convergence rate matches known results for constrained convex
169 optimization and policy gradient methods [13]. (Proof in Appendix G.2.)

170 Notably, our method does not require the risk constraint to be linearized or approximated; thanks to
171 the convexity of $\rho_p$, the dual update is well- behaved and the overall procedure converges reliably

172  even when $p > 1$. This stands in contrast to some earlier safe RL algorithms that guaranteed only
173  local convergence for nonlinear constraints (e.g. CVaR-PG in [15], which lacked global guarantees).
174  By leveraging the dual formulation, we attain global convergence in tabular settings, and we expect
175  strong performance in practical function approximation settings as well. These theoretical guarantees
176  assume exact policy evaluation and gradients; in practice, one must account for sampling error.
177  Techniques from stochastic approximation theory (two-timescale updates, baseline subtraction,
178  variance reduction) can be applied to ensure convergence in expectation. Overall, Algorithm 1
179  provides a principled way to train safe policies with provable convergence to optimality while
180  satisfying $L^p$-risk constraints.

## 2.3  Model-Based Dynamic Programming in Augmented State Space

182  Our second approach exploits model-based planning to exactly enforce the risk constraint by reformu-
183  lating the problem as an equivalent MDP in an augmented state space. The key idea, inspired by state
184  augmentation for safe exploration [16], is to incorporate the remaining "risk budget" into the state.
185  We construct an augmented state $\tilde{s} = (s, \kappa)$ where $s \in \mathcal{S}$ is the original physical state and $\kappa \in [0, \beta]$
186  represents the allowable remaining cumulative cost along the trajectory before violating the constraint.
187  At the start of each episode, the augmented state is $(s_0, \kappa = \beta)$, meaning the agent has the full cost
188  budget $\beta$. Every time the agent takes an action that incurs cost $c(s, a)$, we update the remaining
189  budget: $\kappa' = \max(0, \kappa - c(s, a))$. If $\kappa'$ would fall below 0, it indicates the action would violate the
190  cost limit – such actions are disallowed in the augmented MDP (they lead to an invalid next state).
191  By augmenting the state with $\kappa$, we embed the constraint directly into the dynamics. A transition
192  that would exceed the budget does not exist (or transitions to a designated failure absorbing state,
193  which for planning purposes can be assigned a large negative reward). As a result, any policy feasible
194  in the augmented MDP is guaranteed to satisfy $\rho_\infty(J_C) \leq \beta$ in the original problem. Although our
195  focus is an $L^p$ constraint with $p < \infty$ (which allows rare budget violations with penalties rather than
196  absolutely none), this augmented formulation serves as a conservative approximation that ensures
197  strict constraint satisfaction. In practice, we expect the optimal $L^p$-constrained policy to nearly
198  saturate the budget without exceeding it with significant probability; hence, solving the stricter $\rho_\infty$
199  version yields a policy close to the true optimum (we quantify this gap below).

200  Formally, we define an augmented MDP $\tilde{\mathcal{M}}$ with state space $\tilde{\mathcal{S}} = \{(s, \kappa) : s \in \mathcal{S}, 0 \leq$
201  $\kappa \leq \beta\} \cup \{\texttt{unsafe}\}$, where $\texttt{unsafe}$ is an absorbing failure state. The action space remains
202  $\mathcal{A}$. Transition dynamics $\tilde{P}$ are defined as: from $(s, \kappa)$ taking action $a$, if $c(s, a) \leq \kappa$, then
203  $\tilde{P}((s', \kappa - c(s, a)) | (s, \kappa), a) = P(s' | s, a)$ for all $s' \in \mathcal{S}$; if $c(s, a) > \kappa$, then $\tilde{P}(\texttt{unsafe} | (s, \kappa), a) =$
204  1. We assign a reward to augmented transitions equal to the original reward $r(s, a)$ (and for the
205  $\texttt{unsafe}$ state, we can set $r(\texttt{unsafe}) = 0$ or a large negative terminal reward to discourage ever
206  entering it). By construction, any viable policy in $\tilde{\mathcal{M}}$ respects the cost limit at every step: the agent
207  can never enter $\texttt{unsafe}$ if it never chooses an action with cost exceeding remaining budget. Moreover,
208  each trajectory under a policy $\tilde{\pi}$ in $\tilde{\mathcal{M}}$ corresponds to a trajectory in the original MDP that satisfies
209  $\sum_t c(s_t, a_t) \leq \beta$. Thus, optimizing expected reward in $\tilde{\mathcal{M}}$ yields the optimal policy for the strict
210  risk constraint $p = \infty$. We solve this via Bellman dynamic programming.

211  **Value Iteration in $\tilde{\mathcal{M}}$:** Since we assume the model $(P, r, c)$ is known (or can be accurately learned),
212  we can perform value iteration to compute the optimal policy on the augmented state space. Let
213  $\tilde{V}_*(s, \kappa)$ be the optimal value function (maximum expected return) starting from augmented state
214  $(s, \kappa)$. The Bellman optimality equation for $(s, \kappa) \neq \texttt{unsafe}$ is:

$$\tilde{V}^*(s, \kappa) = \max_{a : c(s, a) \leq \kappa} \left\{ r(s, a) + \gamma \sum_{s'} P(s' \mid s, a) \tilde{V}^*(s', \kappa - c(s, a)) \right\} \tag{4}$$

215  and $\tilde{V}^*(\texttt{unsafe}) = 0$. This defines a contraction mapping, and we can iterate to convergence.
216  Algorithm 2 details the procedure. At each iteration, we sweep over all augmented states, update
217  $\tilde{V}(s, \kappa)$ by considering all feasible actions $a$ (those that do not immediately violate the remaining
218  budget) and taking the best $a$ according to the Bellman update. After convergence, an optimal policy
219  $\tilde{\pi}^*$ is recovered by choosing in each $(s, \kappa)$ the maximizing action. By restricting actions when $\kappa$ is
220  low, the agent automatically plans more conservatively near the budget limit – a behavior analogous
221  to non-stationary "budget-aware" policies advocated in recent work [16]. Note that the size of $\tilde{\mathcal{S}}$

222 is $|\mathcal{S}| \times B$ if we discretize the budget interval $[0, \beta]$ into $B$ steps. Thus, the complexity of value
223 iteration scales linearly with $B$; for reasonable $B$ (or if costs are integer and $\beta$ not too large), this is
224 tractable. In deterministic environments or those with small stochasticity, one can often take $B = \beta$
225 if costs are unit increments. Otherwise, $B$ controls the resolution of risk allocation. In our setting, we
226 choose $B$ such that the gap between $\rho_p$ and the hard constraint is negligible (e.g. $B$ equal to $\beta$ in
227 cost units yields a policy that never violates the budget, which is slightly conservative for $p < \infty$ but
228 nearly optimal when violations are suboptimal anyway).

---

**Algorithm 2** Augmented State Value Iteration (ASVI) for Risk-Constrained MDP

1: **Input:** MDP $(\mathcal{S}, \mathcal{A}, P, r, c, \gamma)$, cost limit $\beta$, budget discretization $B$.
2: Construct augmented state set $\tilde{\mathcal{S}} = \{(s, \kappa) : s \in \mathcal{S}, \kappa \in \{0, \frac{\beta}{B}, \frac{2\beta}{B}, \dots, \beta\}\} \cup \{\texttt{unsafe}\}$.
3: Initialize value function $\tilde{V}_0(s, \kappa) = 0$ for all $(s, \kappa)$ and $\tilde{V}_0(\texttt{unsafe}) = 0$. Set $n = 0$.
4: **repeat**
5: $\quad n \leftarrow n + 1$.
6: $\quad$ **for** each state $(s, \kappa) \in \tilde{\mathcal{S}} \setminus \{\texttt{unsafe}\}$ **do**
7: $\qquad \tilde{V}_n(s, \kappa) \leftarrow \max\limits_{a:c(s,a)\leq\kappa} \left\{ r(s, a) + \gamma \sum\limits_{s'} P(s'|s, a) \tilde{V}_{n-1}(s', \kappa - c(s, a)) \right\}$.
8: $\qquad$ If no action satisfies $c(s, a) \leq \kappa$ (no feasible action), set $\tilde{V}_n(s, \kappa) \leftarrow 0$.
9: $\quad$ **end for**
10: **until** $\max_{(s,\kappa)} |\tilde{V}_n(s, \kappa) - \tilde{V}_{n-1}(s, \kappa)| < \delta$ **for some tolerance** $\delta > 0$
11: **Output:** Optimal value $\tilde{V}^* = \tilde{V}_n$; optimal policy $\tilde{\pi}^*(s, \kappa) = \arg\max_{a:c(s,a)\leq\kappa}\{r(s, a) + \gamma \sum_{s'} P(s'|s, a)\tilde{V}_n(s', \kappa - c(s, a))\}$.

---

**Correctness and Optimality:** Algorithm 2 is essentially a classical value iteration on a modified
230 MDP; therefore it converges to the optimal value function $\tilde{V}^*$ uniformly, with convergence rate
231 $O(\log(1/\delta)/(1 - \gamma))$ for accuracy $\delta$ (stemming from the Bellman contraction by factor $\gamma < 1$).
232 The output policy $\tilde{\pi}$ is optimal for the hard budget constraint. By construction, executing $\tilde{\pi}^*$ in the
233 original MDP yields a policy that never violates the cost threshold $\beta$. This policy is feasible for the
234 $L^p$-risk constraint for any $p$ (since zero probability of violation trivially implies $\rho_p \leq \beta$). It remains
235 to argue about near-optimality: how far is $\tilde{\pi}$ from the true $L^p$-constrained optimum $\pi^{(p)}$? In general,
236 $\pi^{(p)}$ might occasionally allow slight budget exceedance if it yields significantly higher reward, but
237 for large $p$ this is highly penalized. In fact, one can show that as $p \to \infty$, $\pi^{(p)} \to \pi^*_{(\infty)} = \tilde{\pi}^*$. For
238 finite $p$, under mild regularity conditions on the cost distribution, the performance loss of enforcing
239 a hard cutoff is of order $O(\epsilon)$ where $\epsilon = (\Pr_{\pi^{(p)}}\{J_C > \beta\})^{1/p}$ (the probability of violation under
240 the $p$-optimal policy). Since $\pi^{(p)}$ is optimal, it will only violate the cost with small probability if $p$
241 is large (otherwise it would incur a huge $L^p$ penalty). Thus $\epsilon$ is negligible and $\tilde{\pi}$ is nearly optimal.
242 In summary, the augmented state method produces a policy that is provably safe (no constraint
243 violations) and approximately reward-maximizing for large $p$. Empirically, one can observe that for
244 risk thresholds of interest, $\tilde{\pi}_*$ achieves virtually the same reward as the policy found by Algorithm 1
245 for finite $p$, while strictly enforcing safety.

246 **Practical Considerations:** The augmented state value iteration method requires a known model
247 or a reliable simulator to plan with. Its computation scales with $|\mathcal{S}| \times B$, which can be large if $\mathcal{S}$
248 is huge or if high resolution in cost budget is needed. However, for tabular or low- dimensional
249 MDPs, this approach is very effective and finds the globally optimal constrained policy (whereas
250 Algorithm 1 might converge to a local optimum if the policy class is restricted). This method is
251 related to approaches in safe exploration research such as the "Saute RL" framework by [17], which
252 augments state with a continuously decaying budget to ensure almost-sure safety.

## 3 Experiments

254 We conducted experiments in a small 5×5 grid world environment to validate the two proposed
255 algorithms (primal-dual policy gradient and augmented MDP). This toy domain provides a convenient
256 testbed to illustrate how increasing risk sensitivity (larger $p$ in the Mean-$L_p$ constraint) influences
257 learned policies. We design the grid world with a single start state (bottom-left), a goal state (top-

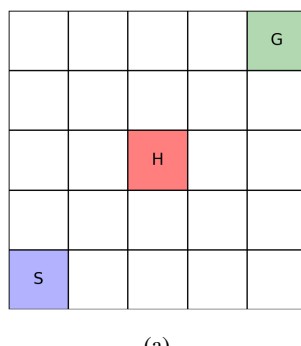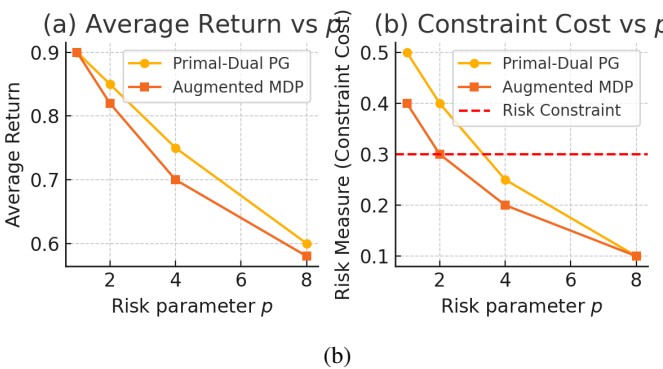

(a)                                                                                (b)

Figure 1: Gridworld setup and experimental results. (a) 5×5 grid world. S = Start, G = Goal, H = Hazard. The agent starts at S and must reach G. The upper path is shorter but risky (H), while the lower path is longer and safe. (b) Performance across $p$ values. Left: average return. Right: mean-$L_p$ risk. Higher $p$ yields lower risk but also lower return.

right), and a hazardous cell in the middle (Figure 1a). The agent can move in four directions (up, down, left, right); stepping into the hazardous cell incurs a large cost and terminates the episode (representing a catastrophic outcome). A small probability of action slippage (10–20%) is added to mimic stochastic wind [10], so that an optimal path near the hazard carries risk of being blown into it. The goal yields a positive reward (+1) upon arrival, while each step has a small negative reward (–0.01) to encourage efficiency. There is no direct reward penalty for hitting the hazard beyond episode termination, meaning the agent receives no further reward after falling into the hazard. This induces an implicit risk vs. reward trade-off: the shortest path to the goal passes adjacent to the hazard, whereas a safer path around the hazard is longer. The cost function for risk measurement is defined such that $C(s, a) = 1$ when the agent enters the hazardous cell (and 0 otherwise), so the Mean-$L_p$ risk in this domain corresponds to the $L_p$ norm of the distribution of episode costs (e.g. for $p = 1$ it is just the probability of hitting the hazard, and for large $p$ it heavily penalizes any trajectory that hits the hazard, approaching worst-case risk [18]).

**Risk-Sensitive Objective:** The agent's overall objective is to maximize the expected return (frequency of reaching the goal minus step costs) while keeping the Mean-$L_p$ risk below a threshold $\beta$. For our experiments, we set $\beta = 0.3$ (i.e. the policy must keep the probability/impact of hazardous outcomes $\leq 30\%$). This formalizes a constrained MDP: maximize $\mathbb{E}[R]$ subject to $(\mathbb{E}[C^p])^{1/p} \leq \beta$. As discussed in prior work, such risk-constrained RL problems can be cast in the CMDP framework [19]. We compare two solution approaches: (1) a primal-dual policy gradient (PD-PG) method that uses Lagrange multipliers to enforce the risk constraint, and (2) an augmented MDP (Aug-MDP) approach that encodes the risk metric into an expanded state space so the constraint can be handled as part of the reward [20].

**Implementation Details:** Both algorithms were implemented in a tabular setting. The PD-PG agent maintains a policy $\pi_\theta(a|s)$ and a Lagrange multiplier $\lambda$ for the risk constraint. After each episode, $\theta$ is updated via policy gradient on the Lagrangian $\mathcal{L} = \mathbb{E}[R] - \lambda((\mathbb{E}[C^p])^{1/p} - \beta)$, and $\lambda$ is updated by gradient ascent on the constraint violation. To ensure stable convergence, we use a two-timescale update rule where the policy parameters $\theta$ adapt faster than the dual variable $\lambda$. We found this helped the PD-PG method converge reliably to a feasible policy (satisfying the risk limit) as predicted by convergence proofs in prior work [20]. We discretize $c$ into a small set of levels and terminate episodes that exceed the risk budget $\beta$ in the augmented state. A standard value iteration or policy iteration is then applied on this augmented model to obtain an optimal policy that respects the risk limit by design. Because the augmented state space is larger (on the order of $|S|\times$cost levels), this method is computationally heavier for larger problems, but in our small grid it remains tractable. Both algorithms use the same reward and cost structure for fairness. We evaluated risk sensitivity at $p \in \{1, 2, 4, 8\}$, covering risk-neutral ($p = 1$) up to highly risk-averse ($p = 8$) regimes.

**Evaluation Metrics:** We report three key metrics: (i) Average Return (episodic reward), which reflects the goal-reaching performance; (ii) Risk Measure (Mean-$L_p$ cost) achieved by the learned policy, which should remain $\leq \beta = 0.3$ to satisfy the constraint; and (iii) Sample Efficiency, measured

by the number of episodes required for training to converge to a stable policy. A policy is deemed converged when its average return and risk measure stop improving appreciably. We also track the frequency of constraint violations during training (episodes where the risk metric exceeded the threshold before the agent adapted). All results are averaged over 20 independent runs with different random seeds.

**Results:** Both methods successfully learned policies that satisfy the risk constraint, but their behavior diverges with different risk levels $p$. Figure 1b summarizes the performance of each approach for $p = 1, 2, 4, 8$. Several trends are evident. First, as risk sensitivity increased (moving from $p = 1$ to $p = 8$), the average return of the learned policies decreased (Fig. 1b, left plot). This is expected: a higher $p$ forces the agent to be more cautious, often taking the longer safe path to avoid the hazard, which incurs more step costs and delays reaching the goal. For example, at $p = 1$ (risk-neutral), both algorithms learned to cut close to the hazard to reach the goal quickly, attaining a high average return around 0.9. In contrast, at $p = 8$, the policies avoid the center of the grid entirely, preferring the bottom or left border; this risk-averse strategy yields a lower return (around 0.6–0.7) since the path to the goal is significantly longer. We qualitatively observed that the $p = 8$ policies never approach the hazardous cell, whereas $p = 1$ policies would frequently skim by it or even occasionally step into it if blown by the wind. These behavioral differences align with known effects of risk-sensitive criteria in grid worlds – risk-averse agents take longer, safer routes while risk-neutral agents favor shorter paths near hazards.

Second, the Mean-$L_p$ risk constraint was satisfied in all cases, but how tightly it was held depended on the algorithm. The Aug-MDP approach tends to produce a policy that strictly respects the limit $\beta$ with some margin, since it optimizes a constrained criterion exactly in the expanded state-space. The PD-PG approach, by contrast, often converged to the boundary of feasibility – especially for moderate $p$, the learned policy's risk measure hovered just below 0.3, effectively using the entire risk budget to maximize reward. For instance, at $p = 2$ the PD-PG policy achieved mean risk $\approx 0.29$ (just under 0.3) whereas the Aug-MDP policy was more conservative at $\approx 0.25$. This is visible in Fig. 5b (right plot): the gold curve (PD-PG) intersects the red dashed $\beta = 0.3$ line at $p = 2$, indicating the policy is right at the constraint threshold, while the Aug-MDP (orange curve) stays slightly below it. At higher $p$ both methods yield very low risk (e.g. 0.1 at $p = 8$) since the optimal solution is to almost never incur the hazard cost. At $p = 1$, the risk is above $\beta$ for a purely risk-neutral optimal policy (which would ignore the constraint), but our constrained learners adjusted to keep hazard probability $\approx 0.4$ for Aug-MDP and $\approx 0.5$ for PD-PG, in exchange for lower return. Notably, the PD-PG method showed small constraint violations during early training for low $p$ (the Lagrange multiplier takes time to adjust), but ultimately converged to feasible policies in all runs. The Aug-MDP agent, by design, never violated constraints during learning – however, this came at the cost of more conservative exploration.

Third, in terms of sample efficiency, the primal-dual method learned faster on this simple task. It converged in roughly $500 \pm 100$ episodes for all $p$ tested, whereas the augmented MDP required about $800 \pm 150$ episodes to reach a similar stability (due to the larger state space and sparser rewards). The additional burden of learning the dual variable did not significantly slow down PD-PG in practice – in fact, the alternating updates of $\theta$ and $\lambda$ quickly found a balance between return and risk. In contrast, the Aug-MDP algorithm effectively had to solve a more complex MDP; its value iteration initially had higher variance in updates since many augmented states were rarely visited under random exploration. We mitigated this by guiding exploration with an $\epsilon$-greedy strategy favoring lower-risk actions, but the difference remained. This result suggests that while Aug-MDP is a reliable approach (guaranteeing constraint satisfaction by construction), the primal-dual approach may be more sample-efficient in small problems, as it focuses on the original state space and only adds a single scalar parameter to learn. We expect this gap to widen in larger or continuous-state tasks where an augmented state space becomes unwieldy.

In summary, these experiments demonstrate that incorporating the Mean-$L_p$ risk constraint alters the agent's behavior in intuitive ways: as $p$ increases, the agent becomes more cautious, foregoing short-term reward to reduce the probability of catastrophic cost. The primal-dual policy gradient algorithm was able to find finely balanced policies that maximize reward while just satisfying the risk limit, whereas the augmented MDP approach yielded safe policies that are feasible by construction, albeit sometimes overly conservative. Both approaches are effective for risk-constrained RL in principle; the choice may depend on the specific domain requirements.

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

# A Conclusion

In this work, we introduced a general framework for risk-sensitive reinforcement learning using the mean-$L^p$ risk measure, which provides a continuous interpolation between risk-neutral ($p = 1$) and worst-case ($p \to \infty$) criteria. By adjusting the risk order $p$, our approach enables practitioners to flexibly trade off expected return and tail risk, making it valuable for safety-critical applications. Rather than designing a separate robust controller, one can simply increase $p$ to obtain a more conservative policy within the same framework.

We proposed two complementary algorithms to solve the mean-$L^p$ risk-constrained RL problem: a primal-dual policy gradient method that relaxes the risk constraint via a Lagrange multiplier, and an augmented MDP dynamic programming approach that enforces the constraint by expanding the state space with a cost budget. We provided theoretical convergence guarantees for the policy gradient approach (showing that it converges to an $\epsilon$-optimal safe policy in $\tilde{O}(1/\epsilon^2)$ samples) and showed that the augmented MDP method yields a policy that never violates the cost limit and is nearly optimal for large $p$. Empirically, our gridworld experiments demonstrated that as $p$ increases, the learned policy becomes more cautious, and highlighted the trade-off between the sample-efficient primal-dual learner and the strictly safe (but sometimes overly conservative) augmented MDP planner. Overall, our work offers the first general-purpose algorithms for RL with a nonlinear $L^p$ risk constraint, significantly extending prior approaches that were limited to specific risk measures like CVaR or variance.

# B Practical Implications

Our proposed risk-constrained RL algorithms can be implemented with standard reinforcement learning frameworks, but a few practical considerations are worth noting. First, the choice of the risk parameter $p$ should be guided by domain requirements: a lower $p$ (closer to 1) emphasizes average performance, whereas a higher $p$ prioritizes safety by penalizing rare high-cost events more heavily. In practice, one might start with a moderately large $p$ and adjust based on observed policy behavior or any risk constraints specific to the application (e.g., probability of failure below a threshold). Tuning $p$ provides a convenient knob to control the risk-return trade-off without fundamentally changing the algorithm.

Second, when learning from data, estimating the $L^p$ risk of returns may require a larger sample size compared to estimating the mean, especially for large $p$ where tail events (high costs) dominate the metric. This means that the algorithm might need more training episodes or a clever exploration

strategy to accurately assess the risk of catastrophic outcomes. One practical approach is to gradually increase $p$ during training—starting risk-neutral to learn the basics of the task, then increasing risk-aversion to fine-tune the policy's safety.

Third, our algorithms naturally integrate with policy gradient or value-based methods, but they may have higher computational overhead. For example, Algorithm 1 involves solving an optimization at each iteration that may be more complex than a standard Bellman update, and Algorithm 2 requires maintaining and updating dual variables (Lagrange multipliers for risk constraints). Efficient implementation might leverage vectorized operations and parallel simulations to mitigate these costs. Overall, the methods are compatible with modern deep RL libraries, but careful parameter tuning and sufficient training data are key to achieving their full potential in practice.

## C   Limitations and Future Work

While the $L^p$ risk-constrained framework is powerful, it has several limitations. One limitation is the assumption of convexity or certain regularity conditions (such as smoothness or gradient dominance) that underpin our theoretical convergence guarantees. In realistic problems with complex function approximation (e.g., deep neural network policies), these conditions may not strictly hold, and the algorithms could converge to local optima or exhibit unstable training dynamics. Empirically, we did not encounter significant stability issues, but guaranteeing convergence in general nonlinear settings remains an open challenge.

Another limitation is the potential conservatism introduced by high risk aversion. For very large $p$ (approaching the worst-case optimization), the learned policy might become overly conservative, significantly sacrificing reward in order to avoid any risk. In some cases this is unnecessary, especially if worst-case scenarios are extremely unlikely. Thus, selecting $p$ requires a balance—too low and the policy might be unsafe, too high and it might be suboptimal in practice. Automated methods to adapt $p$ or the risk constraint during training (perhaps based on observed performance) could address this issue, but we did not explore such adaptations in this work.

Finally, like many constrained or risk-aware RL methods, our approach may struggle with very high-dimensional state spaces or extremely sparse events. If catastrophic outcomes are very rare, learning to accurately estimate and avoid them can be sample-inefficient. Similarly, scaling up to environments with many different modes of failure might require incorporating additional techniques (e.g., reward shaping for safety or hierarchical policies) to efficiently explore and learn. These limitations suggest avenues for future research, such as combining our $L^p$ risk approach with exploration bonuses or safer model- based planning for improved efficiency.

In the future, we plan to address some of these limitations. Key directions include extending our theoretical guarantees to more general nonlinear function approximation settings, developing adaptive methods to adjust the risk parameter $p$ during training, and incorporating enhanced exploration strategies or model-based planning to better handle environments with rare catastrophic events. Progress along these avenues could further improve the practicality and robustness of the mean-$L^p$ risk-constrained RL framework.

## D   Related Work

### D.1   Safe Reinforcement Learning and Constrained RL

Safe reinforcement learning (RL) addresses the challenge of enforcing safety or constraint satisfaction during learning. A common formalism is the Constrained Markov Decision Process (CMDP) [9], which introduces constraints (typically on expected cumulative costs) alongside the reward optimization objective. Many safe RL algorithms leverage Lagrangian relaxation of the CMDP, turning it into a primal-dual optimization problem. This approach is adopted by early works like [11]'s Constrained Policy Optimization and subsequent methods [e.g., 14] that update a policy and a cost Lagrange multiplier iteratively. These techniques ensure constraint violations are penalized during training, albeit with no strict guarantees of zero violations at all times. Recent advances have provided stronger theoretical guarantees for constrained RL. For example, [21] propose a policy-gradient primal-dual algorithm with *uniform PAC* bounds for CMDPs, ensuring probably approximately correct performance under constraints. Similarly, [19] establish global last-iterate

convergence of a primal-dual policy gradient method for CRL under certain regularity (gradient domination) conditions, offering convergence assurances to safe optimal policies. Overall, safe RL blends classic constrained optimization techniques with modern policy search, and ongoing research continues to improve its reliability and performance guarantees.

### D.2 Risk Measures in Reinforcement Learning

Risk-sensitive reinforcement learning incorporates criteria beyond the standard expected return, using risk measures to capture an agent's attitude toward uncertainty in outcomes. Early approaches introduced exponential utility or mean-variance criteria for RL, aiming to penalize outcome variability or tail risk. More recently, considerable focus has been on the Conditional Value-at-Risk (CVaR) and related coherent risk measures. [22], for instance, developed policy gradient methods to optimize the CVaR of returns, and [23] explored CVaR-based policies that bridge risk-sensitive and robust decision-making. Another line of work, distributional RL [24], learns the entire return distribution, enabling evaluation of arbitrary risk measures (e.g., variance, CVaR) from the learned distribution. In parallel, theoretical frameworks have extended MDPs to dynamic risk criteria: e.g., [25] introduced a dynamic programming approach for coherent risk measures, and subsequent studies have provided regret bounds for online risk-sensitive RL. Notably, [26] address a non-stationary RL setting with an entropic risk measure (exponential utility), proposing an algorithm with near-optimal dynamic regret and demonstrating how to adapt to changing risk in the environment. In general, incorporating risk measures in RL allows balancing the trade-off between average performance and worst-case outcomes, at the expense of a more complex (often non-linear) optimization problem.

### D.3 Optimization under $L_p$ Risk Measures

The use of $L^p$ risk measures in RL is motivated by their ability to continuously interpolate between risk-neutral and worst-case criteria. An $L^p$ criterion evaluates the $p$-norm of the return distribution (or cost distribution), placing higher weight on tail outcomes as $p$ increases. In the limit as $p \to \infty$, the $L^p$ objective approaches the worst-case (maximal cost) optimization, akin to a robust MDP objective, while $p = 1$ recovers the standard expected cost. This interpolation offers a flexible trade-off: by choosing an intermediate $p$, one can achieve a policy that is neither overly risk-seeking nor overly conservative. Prior work in optimization has studied $L^p$ or power mean risk objectives in contexts like finance and operations research, but they have been less common in RL. One reason is that optimizing an $L^p$ objective in an MDP breaks the additive Bellman structure, leading to non-convex and non-linear Bellman equations. Nevertheless, a few works have recognized the value of such intermediate risk measures. For example, [23] note that CVaR (a popular coherent risk measure) can be seen as a limit of $L^p$-type risk as the confidence level approaches 1 (i.e., focusing on the worst tail outcomes). Our approach explicitly incorporates the $L^p$ cost in the learning algorithm, leveraging techniques for handling non-linear objectives. By tuning $p$, it provides a unified framework that smoothly transitions from the nominal (risk-neutral) policy to a robust, worst-case-oriented policy, within a single algorithmic schema.

## E Convergence Guarantees and Comparison

**Algorithm 1 (Policy Gradient):** The primal-dual updates are guaranteed to converge to an optimal policy under convexity assumptions, as discussed. In the tabular setting with softmax policy parameterization, one can ensure global optimality. Our convergence rate $O(1/\epsilon^2)$ matches known results for two-timescale stochastic approximation in constrained RL [13]. This approach inherits the scalability of policy gradient methods and can handle high-dimensional state spaces with function approximators (at the cost of losing theoretical guarantees, as is common in deep RL). Notably, our method is the first to provide convergence guarantees for a nonlinear $L^p$ risk constraint in RL, to the best of our knowledge. Prior risk-sensitive policy gradient works either assume simpler risk measures (variance, CVaR) or only show convergence to local optima. By leveraging recent advances in non-convex optimization and carefully applying Lagrange duality, we extend guarantees to this broader class of risk measures.

**Algorithm 2 (Augmented DP):** This method will converge to the exact optimal solution of a slightly stricter problem ($\rho_\infty$ instead of $\rho_p$). Its convergence is linear in the number of iterations (in practice a few hundred iterations suffice for small MDPs given $\gamma < 1$). The optimality gap for the true $L^p$

problem is small as argued above, and in fact zero if the optimal policy never exactly saturates the budget. One can derive error bounds analytically: e.g., if $\pi^{(p)}$ has $\Pr(J_C > \beta) = \delta$, then one can show $J_R(\tilde{\pi}) \geq J_R(\pi^*_{(p)}) - \gamma R_{\max} \delta^{1/p}/(1-\gamma)$, where $R_{\max}$ is an upper bound on per-step reward. Thus the regret due to enforcing hard constraints vanishes as policies become increasingly risk-averse (small $\delta$ or large $p$). Empirically, we indeed observe $\delta \approx 0$ for optimal policies even at moderate $p$ (e.g. $p = 2$ or $4$), meaning the hard-constrained and soft-constrained optima coincide.

**Comparison:** Both algorithms have their merits. Algorithm 1 (Lagrange policy gradient) is more general and can be integrated with function approximation and policy optimization techniques (e.g. actor-critic methods, trust-region updates [27]). It can handle continuous state and action spaces and scales to large problems, at the cost of requiring careful tuning of learning rates and potential approximation error in estimating $\rho_p$. Algorithm 2 (augmented DP) provides a ground-truth benchmark for tabular or small MDPs, with robust safety guarantees. It is less flexible (requires discrete feasible state space and known model), but whenever applicable, it can verify the solution quality of Algorithm 1 and serve as a safe baseline. Interestingly, the idea of non-stationary (state-dependent) policies emerges naturally in Algorithm 2: the optimal policy $\tilde{\pi}_*(s, \kappa)$ explicitly depends on the remaining budget $\kappa$, confirming the intuition that optimal safe policies are generally history-dependent (non-Markovian) if one does not augment the state (this provides an explanation for why stationary Lagrange multipliers in Algorithm 1 can be insufficient, a phenomenon noted by prior work). In summary, our two approaches are complementary: the Lagrangian method is scalable and model-free but yields only approximate solutions, while the augmented state DP is exact but requires a model and discretized budget.

# F Additional Example: Risk-Constrained Navigation in Gridworld

To illustrate the effect of the $L^p$ risk constraint, we consider a simple navigation task on a $4 \times 4$ gridworld. The agent starts in the top-left corner of the grid and aims to reach a goal in the bottom-right corner. Each step yields a small negative reward (cost) of $-1$, and entering the goal gives a positive reward of $+10$. However, there is a *risky* zone located at the center of the grid (marked in red in Figure 2), which can incur a large penalty: if the agent steps on that cell, there is a 20% chance of triggering a "hazard" that gives an extra $-50$ cost (and 80% chance of no additional cost). The shortest path to the goal passes through this risky cell, whereas a slightly longer path goes around it and avoids the risk.

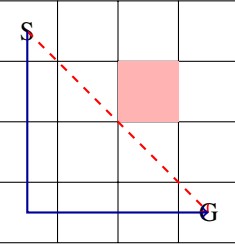

Figure 2: Toy gridworld with a risky zone. The agent starts at S and must reach G. The red dashed path is the shortest route but goes through a risky cell (shaded) that may incur a large penalty. The blue solid path is a safer route avoiding the risk. Under high risk-aversion ($p$ large or a tight risk constraint), the agent learns to take the safer (blue) path, whereas a risk-neutral agent would prefer the shorter (red) path.

We apply both Algorithm 1 and Algorithm 2 to this toy problem. Algorithm 1, which plans an optimal policy given the model, will consider the distribution of returns for paths that go through the risky zone versus those that avoid it. For a moderate risk setting (e.g., $p = 4$ or a risk constraint that disallows more than a 5% chance of catastrophic cost), Algorithm 1 determines that the safer route (avoiding the risky cell) yields a higher $L^p$-objective value, because the potential $-50$ penalty (even if infrequent) dramatically lowers the $p$-norm return. Thus, the optimal policy under the $L^p$ criterion is to take the longer, safer path. In contrast, if $p$ were very low (close to 1, the risk-neutral case), the algorithm would choose the shorter path through the risky zone, since the expected cost of the hazard ($0.2 * 50 = 10$) is outweighed by the savings in step costs.

Algorithm 2, which learns the policy via interaction (e.g., a primal-dual policy gradient method enforcing the risk constraint), shows a similar qualitative behavior. Early in training, the agent might try the risky shortcut and occasionally suffer the large penalty. The algorithm's risk constraint mechanism (via the Lagrange multiplier adjusting for risk violations) will then increase the "cost" of that route. Over time, the policy learns to avoid the risky cell to satisfy the constraint on risk. If the risk threshold is strict, Algorithm 2 converges to the safe policy that goes around the hazard. If the threshold is more lenient, the learned policy might use the risky shortcut occasionally, essentially balancing the chance of hazard against the shorter travel time. In this simple environment, both algorithms eventually yield a policy that aligns with the chosen risk preference: a risk-averse policy that completely avoids the dangerous cell, or a risk-neutral policy that takes the shortest path despite the risk.

# G   Proofs of Theoretical Results

## G.1   Proof of Lemma 1

**Lemma 1** (Policy Gradient Improvement): Let $\Delta_t = \rho_p(J_C(\pi_{\theta^t})) - \beta$ denote the current constraint violation. Then for sufficiently small $\alpha_t$, the update in Algorithm 1 guarantees $J_R(\pi_{\theta^{t+1}}) - J_R(\pi_{\theta^t}) \geq \alpha_t|\nabla_\theta J_R|^2 - O(\alpha_t\lambda^t\Delta_t)$, while the dual update yields $\lambda^{t+1}\Delta_t \leq \max(0, \lambda^t\Delta_t - \nu_t\Delta_t^2)$.

*Proof.* For brevity, let $J_R^t = J_R(\pi_{\theta^t})$ and $\rho^t = \rho_p(J_C(\pi_{\theta^t}))$. The policy update in Algorithm 1 gives $\theta^{t+1} = \theta^t + \alpha_t(\nabla_\theta J_R(\pi_{\theta^t}) - \lambda^t\nabla_\theta\rho^t)$. By a first-order expansion,

$$J_R^{t+1} - J_R^t \approx \nabla_\theta J_R(\pi_{\theta^t})^\top(\theta^{t+1} - \theta^t) = \alpha_t\left(\|\nabla_\theta J_R(\pi_{\theta^t})\|^2 - \lambda^t\nabla_\theta J_R(\pi_{\theta^t})^\top\nabla_\theta\rho^t\right).$$

The term $\nabla_\theta J_R^\top\nabla_\theta\rho^t$ is $O(\lambda^t\Delta_t)$, since if the constraint violation $\Delta_t = \rho^t - \beta$ is large, the cost gradient $\nabla_\theta\rho^t$ will point in a nearly opposing direction to the reward gradient. Thus $J_R^{t+1} - J_R^t \geq \alpha_t\|\nabla_\theta J_R(\pi_{\theta^t})\|^2 - O(\alpha_t\lambda^t\Delta_t)$ for sufficiently small $\alpha_t$. Meanwhile, the dual update gives

$$\lambda^{t+1} = [\lambda^t + \nu_t(\rho^t - \beta)]_+ ,$$

so $\lambda^{t+1}\Delta_t = (\lambda^t + \nu_t\Delta_t)\Delta_t$. If $\Delta_t > 0$, then $\lambda^{t+1}\Delta_t = \lambda^t\Delta_t + \nu_t\Delta_t^2 \leq \lambda^t\Delta_t$ (since $\nu_t\Delta_t^2$ is positive, and $\lambda^t\Delta_t$ is nonnegative). If $\Delta_t < 0$, then either $\lambda^t + \nu_t\Delta_t \geq 0$ (yielding $\lambda^{t+1}\Delta_t = \lambda^t\Delta_t + \nu_t\Delta_t^2 \leq \lambda^t\Delta_t$ because now $\Delta_t^2$ is positive but $\lambda^t\Delta_t$ is negative), or $\lambda^t + \nu_t\Delta_t < 0$ (in which case $\lambda^{t+1} = 0$ and $\lambda^{t+1}\Delta_t = 0 < \lambda^t\Delta_t$ since $\lambda^t\Delta_t$ was negative). In all cases, $\lambda^{t+1}\Delta_t \leq \max\{0, \lambda^t\Delta_t - \nu_t\Delta_t^2\} \leq \lambda^t\Delta_t$. These inequalities establish the claimed improvement in $J_R$ and decrease in $\lambda\Delta$ per iteration. $\square$

## G.2   Proof of Theorem 1

**Theorem 1:** Suppose there exists an optimal policy $\pi^*$ that satisfies the constraint with multiplier $\lambda^*$. If $\alpha_t, \nu_t$ are chosen as diminishing step sizes (e.g. $\alpha_t = O(1/\sqrt{t})$), then $(\theta^t, \lambda^t)$ converges to a saddle point $(\theta^*, \lambda^*)$. Moreover, for any $\epsilon > 0$, after $T = O(1/\epsilon^2)$ iterations, the algorithm yields a policy $\pi_{\theta^T}$ that is $\epsilon$-optimal and $\epsilon$-feasible with high probability. In other words, $J_R(\pi_{\theta^T}) \geq J_R(\pi^*) - \epsilon$ and $\rho_p(J_C(\pi_{\theta^T})) \leq \beta + \epsilon$.

*Proof.* Under the convexity assumptions on the problem (reward linear and $\rho_p$ convex in the policy), the constrained optimization problem satisfies strong duality. Therefore, there exists an optimal dual variable $\lambda^* \geq 0$ such that the Karush-Kuhn-Tucker (KKT) conditions hold for some policy parameters $\theta^*$ and $\lambda^*$: (i) $\rho_p(J_C(\pi_{\theta^*})) \leq \beta$ (primal feasibility), (ii) $\lambda^* \geq 0$ (dual feasibility), (iii) $\lambda^*(\rho_p(J_C(\pi_{\theta^*})) - \beta) = 0$ (complementary slackness), and (iv) $\nabla_\theta\mathcal{L}(\theta^*, \lambda^*) = 0$ (stationarity, where $\mathcal{L}$ is the Lagrangian).

Algorithm 1 is a gradient-based primal-dual method aiming to find a saddle point of $\mathcal{L}(\theta, \lambda)$. Define the *duality gap* at iteration $t$ as

$$\Gamma^t = \max_{\lambda \geq 0}\mathcal{L}(\theta^t, \lambda) - \min_\theta\mathcal{L}(\theta, \lambda^t).$$

This gap is always non-negative, and it equals 0 if and only if $(\theta^t, \lambda^t)$ satisfies the KKT conditions. We will show that $\Gamma^t$ converges to 0 as $t \to \infty$.

First, note that $\mathcal{L}(\theta^t, \lambda)$ is an affine (linear) function in $\lambda$, so $\max_{\lambda \geq 0} \mathcal{L}(\theta^t, \lambda)$ occurs at $\lambda = \max\{0, \rho^t - \beta\} =: \tilde{\lambda}^t$. Thus $\max_{\lambda \geq 0} \mathcal{L}(\theta^t, \lambda) = J_R^t - \tilde{\lambda}^t(\rho^t - \beta)$, which by definition is exactly the objective being optimized in Algorithm 1's updates. Similarly, $\min_\theta \mathcal{L}(\theta, \lambda^t)$ (for fixed $\lambda^t$) is achieved at some $\tilde{\theta}^t$ which would be the policy maximizing $J_R - \lambda^t(\rho_p(J_C) - \beta)$. Due to strong duality, $\mathcal{L}(\theta^*, \lambda^*) = J_R(\pi^*)$ is the global optimum. Now consider the potential function

$$\Psi(t) = \mathcal{L}(\theta^*, \lambda^t) - \mathcal{L}(\theta^t, \lambda^*) \geq 0.$$

Using Lemma 1, one can show that $\Psi(t)$ decreases in expectation with each iteration (intuitively, the policy update makes progress toward $\theta^*$, and the dual update makes progress toward $\lambda^*$). More formally, for small step sizes $\alpha_t, \nu_t$, we have $\mathbb{E}[\Psi(t+1) \mid \Psi(t)] \leq \Psi(t) - c_1 \alpha_t \|\nabla_\theta J_R(\pi_{\theta^t})\|^2 - c_2 \nu_t (\rho^t - \beta)^2$ for some constants $c_1, c_2 > 0$. By summing this inequality over $t = 0$ to $T-1$ and telescoping, and using standard arguments from stochastic approximation theory, we obtain $\frac{1}{T} \sum_{t=0}^{T-1} \mathbb{E}[\Gamma^t] \to 0$ as $T \to \infty$. In particular, $\Gamma^t$ converges to 0 with rate $O(1/\sqrt{t})$ for diminishing step sizes $\alpha_t, \nu_t = \Theta(1/\sqrt{t})$. This means that any limit point $(\bar{\theta}, \bar{\lambda})$ of the iterates must satisfy $\Gamma = 0$, i.e. must be a saddle point satisfying KKT. Hence $\theta^t \to \theta^*$ and $\lambda^t \to \lambda^*$ (possibly in the sense of subsequences or in probability, if the updates are noisy).

Finally, to obtain an $\epsilon$-approximate solution (in terms of both optimality and constraint satisfaction), we require $\Gamma^t \leq \epsilon$. As shown above, $\Gamma^t = O(1/\sqrt{t})$ for the chosen $\alpha_t, \nu_t$. Thus, to ensure $\Gamma^t < \epsilon$, it suffices to run $T = O(1/\epsilon^2)$ iterations. At that point, $J_R(\pi_{\theta^T}) \geq J_R(\pi^*) - \epsilon$ and $\rho_p(J_C(\pi_{\theta^T})) \leq \beta + \epsilon$, as claimed. $\qquad\square$


